# Phascinating Phages

**DOI:** 10.3390/microorganisms10071365

**Published:** 2022-07-06

**Authors:** Marek Straka, Martina Dubinová, Adriána Liptáková

**Affiliations:** 1Medical Faculty, Institute of Microbiology, Comenius University in Bratislava, 81108 Bratislava, Slovakia; marek.straka@fmed.uniba.sk (M.S.); martina.dubinova@fmed.uniba.sk (M.D.); 2Department of Microbiology and Virology, Faculty of Natural Sciences, Comenius University in Bratislava, Ilkovičova 6, 84104 Bratislava, Slovakia; 3St. Elizabeth University of Health and Social Science, 81102 Bratislava, Slovakia

**Keywords:** bacteriophages, drug-resistant bacteria, phage therapy, antibiotic therapy

## Abstract

Treatment of infections caused by bacteria has become more complex due to the increasing number of bacterial strains that are resistant to conventional antimicrobial therapy. A highly promising alternative appears to be bacteriophage (phage) therapy, in which natural predators of bacteria, bacteriophages, play a role. Although these viruses were first discovered in 1917, the development of phage therapy was impacted by the discovery of antibiotics, which spread more quickly and effectively in medical practice. Despite this, phage therapy has a long history in Eastern Europe; however, Western countries are currently striving to reintroduce phage therapy as a tool in the fight against diseases caused by drug-resistant bacteria. This review describes phage biology, bacterial and phage competition mechanisms, and the benefits and drawbacks of phage therapy. The results of various laboratory experiments, and clinical cases where phage therapy was administered, are described.

## 1. Introduction

In recent decades, an increasing number of antibiotic-resistant bacterial strains have been reported. Therefore, conventional antibiotic therapy is often ineffective [1,2,3], leading to the search for new treatment options for bacterial infectious diseases. Several possibilities exist to overcome bacterial resistance to antibiotics; for example, the development of new effective antibiotics [4]; the testing of the antimicrobial effect of natural substances, such as herbal products [5,6], honey [7] or wine products [8]; the use of photodynamic inactivation [9] or the quorum-quenching approach [10]; or therapy that modulates the patient’s own immunity using immunomodulators of microbial origin or therapeutic autovaccines prepared from the patient’s own strains [11]. One of the most promising alternatives to antibiotic treatment is bacteriophage therapy, which uses natural enemies of bacteria, bacteriophages [12]. Bacteriophages, or phages, are viruses that control the growth and spread of their bacterial hosts. They are the most widespread entity in the biosphere and are found wherever bacteria live, for example, in salt waters, cold waters, hot springs, waste waters, or sewage [13,14,15]. They are also the most abundant component of the human microbiome, playing an important role in intestinal microbial composition and horizontal gene transfer [16,17,18]. Due to their unique life cycle and biological properties, phages are a potential powerful weapon in the treatment of bacterial infections. These properties, the differences between antibiotics and phages, and the developments in phage therapy are discussed in the following sections.

## 2. Phage Biology

### 2.1. Phage Life Cycle

The first step during phage infection is adsorption of phages to the surface structures of bacteria by tail fibers or bacteriophage spikes [14]. Phages are capable of infecting only those bacteria that have the corresponding receptor, which determines the host spectrum of the phages. This spectrum is also limited by the defense mechanisms of bacteria against phages [18]. This interaction results in a different host specificity—some phages are strain-specific, whereas others are able to infect a wider range of bacterial species or even genera [19].

After successful adsorption, a pore is formed in the membrane of the infected bacterium and the phage genome is subsequently injected into its cytoplasm [13]. Phage replication may be performed in two basic ways [19,20]. In the lytic cycle, phages exploit the proteosynthetic apparatus of bacteria to synthesize their proteins, replicate phage nucleic acid (NA), assemble phage particles, and release themselves from the cell while disrupting the host bacterial cell [14]. During the release, phages use holins to perforate the cytoplasmic membrane and endolysins to destroy the cell wall membrane, thus lysing the bacterial cell [21].

In the lysogenic cycle, the phage genome is integrated into the bacterial chromosome in the form of a prophage. Such an integrated phage can persist in the genetic material of the bacterium for a long time, and replicate its nucleic acid together with the replication of the bacterial chromosome. Prophages may persist in this state until they switch to the lytic cycle [20]. Lysogeny may provide benefits to the bacterial cell because phages are able to construct gene-encoding products that are involved in antibiotic resistance or virulence factors that enhance the infectious process (e.g., genes for diphtheria toxin [22] or Panton–Valentin leukocidin [23]). Therefore, in the context of phage therapy, a lytic life cycle is desirable, in addition to the absence of genetic determinants of antimicrobial resistance and bacterial virulence in the genome of therapeutic phages [15]. Differences between the lytic and lysogenic life cycle of bacteriophages are shown in Figure 1.

### 2.2. Phages versus Bacteria

During evolution, bacterial strains developed mechanisms to protect themselves from phage infection. One of these mechanisms is superinfection exclusion. In this case, the lysogenized bacterium that carries a prophage in its genome cannot be infected by another phage. Other mechanisms include preventing phage adsorption to the bacterial cell surface due to alteration of the receptor [18] or the production of protective surface polysaccharides that mask the receptor [24]. In addition, bacterial cells can reduce the number of potentially infectious bacteriophage particles by producing membrane vesicles [25]. If phage adsorption is successful, the host cell can subsequently prevent injection of phage NA by modification of the inner membrane proteins [26]. The infected bacterium can also identify the injected phage NA and degrade it via a restriction–modification system [27] or a Clustered Regularly Interspaced Short Palindromic Repeats/CRISPR associated proteins system (CRISPR-Cas) [28]. Another possibility is preventing the replication of phage NA by secondary metabolites produced by the bacterial cell [29], or preventing the assembly of virions by inhibiting phage terminase or by scattering phage coat proteins [30]. If these bacterial cell defense mechanisms are not successful, the host cell may sacrifice itself for the benefit of others, resulting in cell death, and thus limiting the production of phage particles that may infect other bacterial cells in the environment (Figure 2) [31].

However, bacterial strains rarely have more than one defensive mechanism against phage infection [32]. Alternatively, phages may possess mechanisms that help overcome the defensive machinery of the host cell. These include, for example, the ability to produce glycosidases to degrade host capsules, and thus unmask receptors to initiate phage infection [33]. Another mechanism is the ability of phages to produce hypervariable receptor-binding proteins that allow binding to modified host receptors [18,34]. Phages can also overcome the CRISPR-Cas system by mutating or deleting target sites for this system or expressing proteins that interfere with CRISPR-Cas [35]. An important mechanism is the ability of phages to modify their NA to avoid the host restriction–modification system [36], or even the ability of phages to encode their own CRISPR-Cas system that interferes with host defensive mechanisms [37]. All of these mechanisms determine the specificity of phages against bacterial hosts [12].

## 3. Properties of Phage Therapy

### 3.1. Antibiotic versus Phage Therapy

Phage therapy differs from antibiotic therapy in many ways. As mentioned in the previous section, phages attack a much narrower range of bacterial strains than antibiotics. The administration of antibiotics, especially broad-spectrum antibiotics, can often have adverse effects on the balance of the human microbiota resulting in, for example, post-antibiotic diarrhea [38]. Unlike antibiotic treatment, phage therapy is much gentler on the physiological microbiota [13]. However, this narrow host specificity of therapeutic phages sometimes requires precise knowledge of the infectious agent, which presupposes cultivation of the infection site and verification of sensitivity to commercial phage preparations, or the identification of suitable phages for individualized patient treatment [15]. A wider range of potential infectious agents is covered by polyvalent phage cocktails containing several types of phages having extended host specificity [13,39].

A huge advantage of therapeutic phage preparations over antibiotics is the absence of toxic side effects of phages against mammalian cells. Another advantage of phages is their bactericidal effect, as the lytic cycle results in the destruction of the host cell. Phages kill their host bacterial cells in the last phase of their lytic replication cycle, when new phage particles are released from the bacterium. This also increases the number of phage particles directly at the site of infection, and thus increases the probability of infection by other bacteria present [39].

Since the mechanism of action of phages is different from that of antibiotics, antibiotic resistance mechanisms do not affect phage efficacy. Therefore, bacteriophages may be used in the treatment of infections caused by resistant or multidrug-resistant bacterial strains [12,40].

The great potential of bacteriophages has also been observed during biofilm removal, where their ability to enter the biofilm by gradual lysis of individual bacterial layers is manifested. Biofilm can also be removed by the action of phage depolymerases [41]. An interesting alternative is to use a combination of antibiotics with phages [39].

Similarly, like all microorganisms, phages are also able to elicit an immune response in the human body. Detection of phages can occur through pattern recognition receptors. Subsequently, the phagocytes are activated and begin to penetrate the site of infection, which can also contribute to the elimination of bacterial agents [42].

Disadvantages of phage therapy include the potential formation of neutralizing antibodies, which could reduce the effectiveness of phages during their longer administration or repeated phage therapy with the same phages. Prevention of such phage “disposal” can be achieved by improving dosing regimens or by exchanging the phages used in therapy [12]. Another possible risk is that phage preparations may contain endotoxins released from the cell wall of host bacteria; this can be completely eliminated by appropriate phage suspension technology and efficient purification methods [16]. However, the release of endotoxins can also occur during the lysis of bacterial cells by phages in the patient’s body, which can trigger a cascade of immune responses and subsequent adverse events upon systemic administration. However, similar reactions may also accompany antibiotic therapy [39]. Due to the fact that phage preparations are predominantly administered locally and, more rarely, systemically, the risk of such reactions is very low [2,3,43,44,45,46,47,48].

### 3.2. Phage Administration

Phage preparations may be administered topically, orally, by aerosol, in the form of suppositories, intravenously, intraperitoneally, intramuscularly, or subcutaneously. The simplest and most widely used method of administration in clinical practice is the local application of a phage preparation, due to which it is possible to achieve high concentrations of phages at the site of infection. A limitation of this procedure is the risk of leaching the phage preparation from the application site, which can be avoided by using phages incorporated into the gel carrier or in the emulsion. The advantage of oral use of phage preparations is the possibility of using higher doses of the preparation. The disadvantage is that the acidic environment of the stomach can reduce the number of active phages [12]. The solution may be the use of phages on an empty stomach after neutralization of the stomach content with sodium bicarbonate [49], or in the form of microcapsules in which the phages are protected by biopolymers resistant to gastric acid and intestinal juice [50]. Phages can reach areas that are poorly congested in the form of aerosol, but the disadvantage is that there are high losses in phage concentration due to the presence of mucus. Better penetration of phage preparations through the mucus layer can be ensured by using depolymerases. For intravenous administration, the rapid systemic diffusion of phage particles is an advantage, but there is a danger of their rapid removal by neutralizing antibodies. The solution may be the selection of low-immunogenic phages. In the case of intramuscular and subcutaneous administration of phage preparations, phages are delivered directly to the site of infection, but only low concentrations are achieved. This limitation may be resolved by increasing the number of doses [12]. A comparison of the properties of phage and antibiotic therapies is summarized in Table 1.

## 4. Phage Therapy at Present

Despite the fact that bacteriophages were discovered as early as 1917, the use of phage therapy was suppressed globally by the discovery and development of antibiotics. Phage therapy has a rich tradition in some countries of Eastern and Central Europe, especially in Georgia, Russia, and Poland. The Eliava Institute in Tbilisi [51] and Microgen in Moscow [52] are best known for their production of bacteriophage preparations. These institutions focus on the production of various commercial phage cocktails such as Bakteriofag-Stafilokokovyj, Sextaphag^®^ (both NPO Microgen), Staphylococcal Bacteriophage, Fersisi Bacteriophage, SES-Bacteriophage, Intesti-Bacteriophage, Enko-Bacteriophage, and Pyo-bacteriophage (Eliava BioPreparations). Generally, they are recommended for local usage in patients with pyogenic infections, such as infections of skin and soft tissues, infections of the respiratory tract, infections of the eye, urinary tract infections, or decolonization of infectious agent carriers in the nasal cavity. Intesti-Bacteriophage and Enko-Bacteriophage are focused mainly on the treatment of gastrointestinal infections [53,54,55,56,57,58,59,60].

The Institute of Immunology and Experimental Therapy in Wroclaw [61] deals with phage therapy to a significant extent. In the Czech and Slovak republic, a STAFAL^®^ phage preparation having an antistaphylococcal effect is registered and indicated for local use [62]. However, phage therapy is still not widely used in most countries due to the lack of efficacy benchmarks and well-established information on safety, approved manufacturing practices, and standard protocols for the treatment [12,63]. In connection with the development of bacterial resistance to antibiotics, efforts to produce safe and effective phage preparations have also been made in the West [64].

### 4.1. Phage Laboratory Studies

Despite the limited use of phage preparations, many have already been investigated in several in vitro and in vivo studies [65,66,67,68,69,70].

The effect of the antistaphylococcal preparation STAFAL^®^ was demonstrated in the work of Dvořáčková et al., with significant efficacy in methicillin-resistant *Staphylococcus aureus* (MRSA) strains [40]. The high efficacy of Staphylococcal Bacteriophage and Pyo-bacteriophage was also demonstrated in the experimental study against MRSA strains in biofilm [65]. In vitro susceptibility of this bacterium was also proven in the study published by Verstappen; however, ex vivo and in vivo susceptibility was not established [66]. In contrast, phage treatment of mastitis in mouse models was successful [67].

The susceptibility of extended-spectrum beta-lactamase-producing *Escherichia coli* strains to phage preparations, such as SES-Bacteriophage, Intesti-Bacteriophage, Enko-Bacteriophage or Pyo-bacteriophage, was determined by Gundogdu et al. SES-bacteriophage was effective against 59.2% of the tested strains, and all other cocktails were active against more than 80% of the strains [68]. A more recent study described newly isolated phages that work well in resistant uropathogenic *E. coli* strains [69].

Namonyo et al. and Camens et al. discovered novel phages that are active against strains of *Pseudomonas aeruginosa*. Further analysis of the isolated PA4 phage revealed its good stability under various temperature and pH conditions, and the absence of gene-encoding antibiotic resistance mechanisms and the production of toxins [70,71]. In the study published by Fong et al., phages were able to reduce the biofilm formed by *P. aeruginosa* in vitro [72].

Phage efficacy against different mycobacterial species was demonstrated by several laboratory studies (reviewed by [73]).

### 4.2. Phage Case Studies

Several clinical cases in which phage therapy has been administered have also been published [2,3,44,45,46,47,48,74,75,76]. Jennes et al., described the case of a man in his 60s who developed a pressure ulcer during hospitalization that was colonized by a resistant strain of *P. aeruginosa*. Subsequently, the patient developed sepsis and was treated with colistin as the only effective antibiotic. The patient’s condition was complicated by acute renal failure, which led to the need to discontinue colistin. Phages whose efficacy had been demonstrated in vitro were used in experimental therapy. Phage therapy was administered intravenously to the patient. The patient’s kidney function improved within a few days and the collected blood cultures were negative. However, pressure ulcers remained colonized by strains of *P. aeruginosa* and other species of pathogenic microorganisms. Four months later, the patient developed a septic condition due to the presence of a strain of *Klebsiella pneumoniae* in the patient’s blood, leading to cardiac arrest and the death of the patient [2].

In another case, a 15-year-old patient with cystic fibrosis and various comorbid conditions underwent a lung transplant and was affected by disseminated *Mycobacterium abscessus* infection. For the treatment, a phage preparation was designed and applied intravenously; this was well tolerated, lung and liver function improved, and the skin nodes healed after 6 months. Phage-neutralizing antibodies were not detected [45]. Contradictorily, in the case of 81-year-old man infected by *M. abscessus*, intravenous phage treatment failed due to the phage-neutralizing antibody response [46].

In a 7-year-old cystic fibrosis patient infected with *P. aeruginosa* and *S. aureus* strains, there was no significant improvement in clinical condition after antibiotic therapy. Therefore, phage therapy was initiated using Pyo-bacteriophage, which was applied by nebulization. Treatment was administered nine times at intervals of 4–6 weeks and the quantity of the strain of *P. aeruginosa* was significantly reduced, but the preparation of the phage was not effective against the strain of *S. aureus*. Subsequently, bacteriophage Sb-1 was added to Pyo-bacteriophage to eliminate the *S. aureus* strain. The modified phage preparation was administered to the patient five times. The result was a significant reduction in the amount of *S. aureus* strain, with no serious side effects during treatment [74].

Phage therapy has also been used successfully in the treatment of patients with prosthetic joint infection. A 62-year-old diabetic with a history of complete knee arthroplasty overcame several episodes of prosthetic knee infection. Despite numerous surgeries and long-term antibiotic treatment, he was at risk of limb amputation due to persistent periimplantitis of the knee caused by *K. pneumoniae*. However, phage therapy was initiated, administered intravenously in 40 doses. In addition, he continued to receive minocycline due to a previous infection caused by strains of *Enterococcus faecalis and Staphylococcus devriesei/hemolyticus*, whose eradication was unsuccessful. This therapy led to remission of local symptoms, signs of infection, and restoration of knee function. The patient had no treatment-related adverse events and remained healthy for 34 weeks after cessation of treatment while still receiving minocycline [75].

In a double-blind, randomized study, a six-phage cocktail from Biocontrol Limited (BiophagePA) or placebo (glycerol-PBS solution) was applied to the ear canal in 24 patients with otitis media caused by *P. aeruginosa*. Treatment success was monitored after 7, 21, and 43 days, and revealed a statistically significant improvement in clinical status in patients treated with the phage compared to the control. No adverse reactions were reported in the phage group [3].

A 47-year-old patient after head trauma developed an abscess and ventriculitis caused by a multidrug-resistant strain of *Acinetobacter baumanii*. Intravenous phage therapy was initiated and, after 8 days, the patient’s cerebrospinal fluid was culture-negative for *A. baumanii*; however, it was positive for *K. pneumoniae* and *S. aureus*, and the patient died [47].

A very promising application of phages is their use in the treatment of urinary tract infections, due to the occurrence of resistant strains, limited possibilities of antibiotic treatment, and frequent occurrence of relapses in patients. In a study described by Ujmajuridze et al., nine patients who underwent urinary tract surgery and subsequently had a urinary tract infection received Pyo-bacteriophage phage preparation through a suprapubic catheter. The efficacy of this phage preparation was tested in vitro and adapted for the treatment of urinary tract infections (active against *S. aureus, E. coli, Streptococcus* spp., *P. aeruginosa*, and *Proteus mirabilis*). Treatment was administered twice a day for 7 days. In most patients, the pathogen titers decreased significantly and, in addition, no adverse events were reported during treatment [76].

A 60-year-old man with a left ventricular assist device was infected by multidrug-resistant *P. aeruginosa*. Phage preparation containing three phages was applied intravenously four times. The culture became negative and the patient’s condition improved [47].

Within the Phagoburn project [64], Jault et al. tested a phage preparation in the treatment of burns infected with *P. aeruginosa*. However, the efficacy of the preparation was lower compared to that of standard therapy (1% sufadiazine cream emulsion), probably due to insufficient phage titers at the site of infection [44].

Schooley et al. described a 68-year-old man with a pancreatic pseudocyst infected by *A. baumanii*. After unsuccessful antibiotic therapy, phage therapy was administered intravenously, with a marked improvement at 48 h. After 11 weeks of therapy, complete recovery was achieved [48].

## 5. Conclusions

In the post-antibiotic era, interest in phage therapy is increasing because lytic phages are able to destroy bacterial cells, and thus treat infections. Throughout evolution, bacteria have competed for survival by producing defensive mechanisms, and phages have developed methods to overcome these mechanisms. This review describes the advantages and disadvantages of phage therapy, various laboratory studies, and clinical cases where phage therapy has been administered successfully. The importance of phage therapy has been demonstrated, especially in the treatment of infections caused by multidrug-resistant bacterial strains. Therefore, phages can play an important complementary role to antibiotics.

## Figures and Tables

**Figure 1 microorganisms-10-01365-f001:**
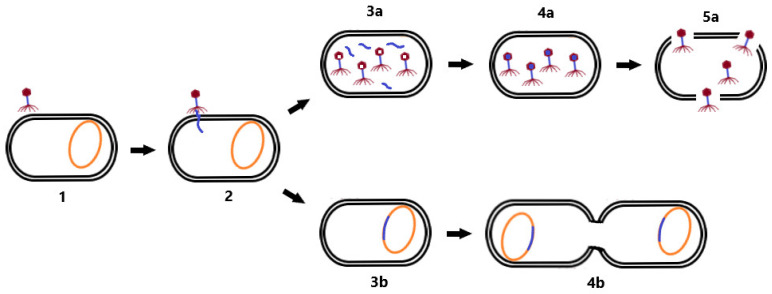
Life cycle of bacteriophages. First, the phage adsorbs to the bacterial cell receptor (**1**). Second, it forms a pore in the membrane and phage NA is injected into the host cell (**2**). As the virus continues the lytic cycle, viral proteins are produced, phage NA (**3a**) replicates, and virion assembly (**4a**) and cell lysis (**5a**) occur. During the lysogenic cycle, the phage integrates its genome into the host chromosome (**3b**) and replicates with it (**4b**). Changing conditions may induce a transition to the lytic cycle.

**Figure 2 microorganisms-10-01365-f002:**
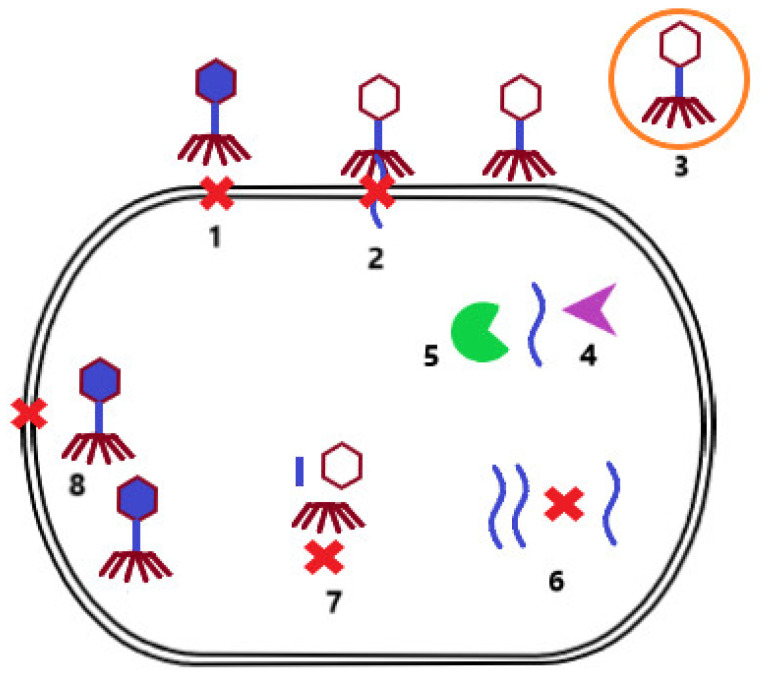
Mechanisms of bacterial resistance to phage infection. Prevention of phage adsorption to the surface of bacterial cells due to alteration of the surface receptor or production of protective surface polysaccharides (**1**). Prevention of phage NA injection by modification of inner membrane proteins (**2**). Reducing the number of free infectious bacteriophage particles by producing membrane vesicles (**3**). Degradation of phage NA by the restriction modification system (**4**) or the CRISP-Cas system (**5**). Prevention of phage NA replication by secondary bacterial metabolites (**6**). Inhibition of the assembly of phage particles by blocking terminase or scattering envelope proteins (**7**). Limitation of phage particle production by induction of cell death (**8**).

**Table 1 microorganisms-10-01365-t001:** Comparison of aspects of phage and antibiotic therapies [12,39].

Property	Phage Therapy	Antibiotic Therapy
**Pharmacokinetics**	Phages replicate at the site of infection; after elimination of the bacterial host, spontaneous disappearance occurs	Antibiotics are metabolized and eliminated by human body
**Spectrum of action**	Phages attack bacteria based on their host specificity	Antibiotics usually have a much broader spectrum of action
**Resistance**	Phages are suitable for the treatment of infections caused also by antibiotic-resistant bacteria; phage-resistant bacterial strains have lower fitness	The number of antibiotic-resistant bacterial strains is increasing
**Manufacture**	Simple isolation of phages and their modification for therapeutic use	Challenging development of new antibiotics with high financial costs
**Effect on biofilm**	Production of depolymerization enzymes for biofilm elimination	Reduced effect on the bacteria in the biofilm
**Clinical validation**	Few clinical studies	Many clinical studies
**Combined therapy**	Efficacy is higher with combined therapy	

## Data Availability

Not applicable.

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
