# Peer review of "Phascinating Phages"

_microorganisms, 2022, doi:10.3390/microorganisms10071365_

Round 1

Reviewer 1 Report

A resurgence of interest in phage therapy has given rise to a proliferation of review articles on phage therapy. This concise review strikes a good balance between the key issues ; the basics of phage lifecycles; mechanisms of bacterial resistance to phage attack; comparison of phage vs antibiotic treatment; a summary of phage based products already available and importantly a summary of recent key case studies in the West.

There are a few examples where phrasing could be improved:

 abstract line 1. Treatment of bacterial infections has become more complex due to the …..

Page 1 line 43. clarify what is meant by flagellar fibres here. Flagella-like fibres? This may cause confusion as written.

Page 2 line 55. In this sentence describe holins before endolysins, as the accepted mechanism is that holins make pores to allow endolysins access to the cell wall.

Page 2 line 64. …a lytic lifecycle is desirable….

Page 4 line 165. Rephrase: …locally and more rarely systemically…. However, several of the examples quoted later use intravenous administration of phage.

Figure 2 could be updated to make it more consistent with the style of figure 1 – specifically the line widths of the components in figure 2 could be reduced.

Author Response

Dear reviewer, 

Thank you very much for the thorough review of our manuscript and for your valuable suggestions. We hope, that we satisfactory clarified all disputable points of our manuscript, what allowed to improve the correctness and comprehensibility of the revised text.  

Please find below our responses: 

Point 1: abstract line 1. Treatment of bacterial infections has become more complex due to the …..

Response 1: We edited the sentence according to your suggestion.

Point 2: Page 1 line 43. clarify what is meant by flagellar fibres here. Flagella-like fibres? This may cause confusion as written.

Response 2: We changed the “flagellar fibers” to “tail fibers” to clarify the statement.

Point 3: Page 2 line 55. In this sentence describe holins before endolysins, as the accepted mechanism is that holins make pores to allow endolysins access to the cell wall.

Response 3: We first described holins and subsequently endolysins.

Point 4: Page 2 line 64. …a lytic lifecycle is desirable….

Response 4: We rephrased the sentence.

Point 5: Page 4 line 165. Rephrase: …locally and more rarely systemically…. However, several of the examples quoted later use intravenous administration of phage.

Response 5: We rephrased the sentence and assigned appropriate references describing intravenous administration of phages.

Point 6: Figure 2 could be updated to make it more consistent with the style of figure 1 – specifically the line widths of the components in figure 2 could be reduced.

Response 6: We edited the figure 2 to be more consistent with style of the figure 1 especially line widths were reduced.

Sincerely yours, 

assoc. prof. Adriána Liptáková, MD, PhD, MPH

Reviewer 2 Report

The review is very informative, the topic is interesting, but I have some comments and objections. The goal of the review is to describe the phascinating phages and the advantages of the phage therapy. Unfortunately in introduction there is given so different information which interferes to focus on main topic, e.g.,  about herbal products, honey, wine, immunomodulators, autovaccines etc. About phages there are two sentences only. Why?

Another objection is about chapter Properties of Phage therapy. I can't agree with your statement that "The most common source of potential therapeutic phages is wastewater". After that you continue that the main criteria for the selection of phages is .... etc. Where from selection? From wastewater? 

These chapters should be revised.

Author Response

Dear reviewer, 

Thank you very much for the thorough review of our manuscript and for your valuable suggestions. We hope, that we satisfactory clarified all disputable points of our manuscript, what allowed to improve the correctness and comprehensibility of the revised text.  

Please find below our responses: 

Point 1: The review is very informative, the topic is interesting, but I have some comments and objections. The goal of the review is to describe the phascinating phages and the advantages of the phage therapy. Unfortunately in introduction there is given so different information which interferes to focus on main topic, e.g.,  about herbal products, honey, wine, immunomodulators, autovaccines etc. About phages there are two sentences only. Why?

Response 1: We revised the introduction to focus more on the main topic, bacteriophages.

Point 2: Another objection is about chapter Properties of Phage therapy. I can't agree with your statement that "The most common source of potential therapeutic phages is wastewater". After that you continue that the main criteria for the selection of phages is .... etc. Where from selection? From wastewater? 

Response 2: We deleted this paragraph to improve the consistency and organization of the text.

Sincerely yours, 

assoc. prof. Adriána Liptáková, MD, PhD, MPH

Round 2

Reviewer 2 Report

Dear authors,

Thank you for your comments and corrections you have made.